# A Validation Study of cT-Categories in the Swedish National Urinary Bladder Cancer Register—Norrland University Hospital

**DOI:** 10.3390/jpm13071163

**Published:** 2023-07-20

**Authors:** Erik Wiberg, Andrés Vega, Victoria Eriksson, Viqar Banday, Johan Svensson, Elisabeth Eriksson, Staffan Jahnson, Amir Sherif

**Affiliations:** 1Department of Surgical and Perioperative Sciences, Urology and Andrology, Umeå University, 907 36 Umea, Sweden; wiberg.erik@outlook.com (E.W.); andve1907@gmail.com (A.V.); victoria.eriksson@umu.se (V.E.); viqar.banday@umu.se (V.B.); 2Department of Statistics, Umea School of Business, Economics and Statistics (USBE), Umea University, 907 36 Umea, Sweden; johan.svensson@umu.se; 3Department of Radiation Sciences, Diagnostic Radiology, Umeå University, 907 36 Umea, Sweden; elisabeth.k.eriksson@regionvasterbotten.se; 4Department of Urology, Biomedical and Clinical Sciences (BKV), Linköping University, 581 83 Linköping, Sweden; staffan.jahnson@gmail.com

**Keywords:** urinary bladder neoplasm, neoplasm staging, validation study, tumour in bladder diverticulum, hydronephrosis

## Abstract

Background: In Sweden, all patients with urinary bladder cancer (UBC) are recorded in the Swedish National Register for Urinary Bladder Cancer (SNRUBC). The purpose of this study was to validate the registered clinical tumour categories (cT-categories) in the SNRUBC for Norrland University Hospital, Sweden, from 2009 to 2020, inclusive. Methods: The medical records of all 295 patients who underwent radical cystectomy for the treatment of UBC were reviewed retrospectively. Possible factors impacting the cT-categories were identified. To optimise cT-classification, computed tomography urography of all patients with suspected tumour-associated hydronephrosis (TAH) or suspected tumour in bladder diverticulum (TIBD) were retrospectively reviewed by a radiologist. Discrepancy was tested with a logistic regression model. Results: cT-categories differed in 87 cases (29.5%). Adjusted logistic regression analysis found TIBD and TAH as significant predictors for incorrect registration; OR = 7.71 (*p* < 0.001), and OR = 17.7, (*p* < 0.001), respectively. In total, 48 patients (68.6%) with TAH and 12 patients (52.2%) with TIBD showed discrepancy regarding the cT-category. Incorrect registration was mostly observed during the years 2009–2012. Conclusion: The study revealed substantial incorrect registration of cT-categories in SNRUBC. A major part of the misclassifications was related to TAH and TIBD. Registration of these variables in the SNRUBC might be considered to improve correct cT-classification.

## 1. Introduction

Urinary bladder cancer (UBC) is one of the most-common cancers in the world and affects men more than women. Urothelial cancer is the most-common subtype of UBC with 98% of cases. Nearly 75% of newly diagnosed UBCs are non-muscle-invasive bladder cancer (NMIBC), while the remaining 25% are muscle-invasive bladder cancer (MIBC) [1,2,3]. Since 1997, all patients diagnosed with UBC are recorded in the Swedish National Register of Urinary Bladder Cancer (SNRUBC). Detailed pre-, peri-, and post-operative data including complications have been registered in a separate form in the SNRUBC since 2011 [4].

The Union for International Cancer Control TNM classification is an internationally accepted cancer staging classification, and this classification was used for newly diagnosed UBC cases [5]. Treatment strategies for UBC are dependent on the clinical tumour classification; therefore, the correct classification is of a paramount consideration [6]. According to the TNM classification, the categories Ta, Tis, and T1 are NMIBC. Categories T2, T3, and T4 are MIBC [1,7]. Clinical tumour staging (cTNM) includes bimanual palpation (BP) before and after transurethral resection of bladder tumour (TURb), cystoscopy and radiographic assessment with respect to local tumour development including tumour-associated hydronephrosis (TAH) and tumour in bladder diverticulum (TIBD), as well as regional lymph node involvement and distant metastases [1]. The preoperative presence of TAH has previously been recognised as a significant marker for advanced disease and a prognostic marker for survival [8]. Two different meta-analyses confirmed that preoperative hydronephrosis was associated with advanced disease and poor survival rate [9,10]. Bladder diverticulum (BD) in adults lacks muscularis propria in its wall and is named “pseudodiverticula” [11]. Therefore, many authors suggest that there is no pT2 in bladder cancer within BD, only pT1 and pT3. UBC arising from diverticula is rare, making up only 0.8–10% of all cases [12]. The diagnosis and management of TIBD are challenging [11,13]. Despite that, Vouskuilen showed that upstaging of tumours in BD was a frequent problem. Many tumours invading lamina propria were staged as ≥pT2 after cystectomy [14].

Urothelial carcinoma may present with variant histologies. These variant histologies have different properties than classical urothelial cancer and are generally associated with more-advanced disease [15]. It is important to classify any presence of variant histology regardless of NMIBC or MIBC for prognostic and treatment evaluation purposes [15,16]. The malignant potential and clinical features of each subtype are still being investigated [17]. The detection of variant histology subtypes is difficult and can vary between different centres [18].

The registration in the SNRUBC is usually performed by highly qualified urological nurses with great experience in urology regarding the registration of urological tumours. Usually, a local urologist provides assistance. However, in some centres, due to a lack of experienced staff, other less-experienced staff members might have performed the registration. In some cases, the assistance from a urologist might have been less accessible. No registration points for TIBD or for TAH exist in the SNRUBC, and whether they are considered regarding the clinical categorisation of the primary tumour (cT-classification) is not known.

In the present study, we investigated the cT-categories registered in the SNRUBC and managed at Norrland University Hospital in Sweden for the years 2009–2020. This analysis was restricted to bladder cancer patients who underwent cystectomies during the period. The primary objective was to determine the total incidence of misclassification of the cT-category. Secondarily, the incidence of misclassification related to TAH and TIBD was investigated.

## 2. Materials and Methods

A retrospective review of the medical records of all patients who underwent cystectomy for urinary bladder cancer for the years of 2009–2020, inclusive, at Norrland University Hospital was performed. Norrland University Hospital is the largest urosurgical centre in the northern healthcare region of Sweden, a region with a population of 650,000 [19]. A complete list of all patients with cystectomies as the primary treatment of UBC was compiled. Localised UBC of urothelial origin, registration in the SNRUBC, and completion of radical cystectomy (RC) were the criteria for inclusion. The final cohort comprised 295 patients.

The validated cT-category was defined as a thorough re-registration regarding the cT-category. This included the review of all patient records regarding cystoscopy, TURb pathological T-category, and radiographic assessment with regard to local tumour status including TAH and TIBD. Decisions on multi-disciplinary conferences and re-resection with TURb were taken into consideration when analysing the clinical tumour classification. However, no complete data regarding the number of patients discussed in multi-disciplinary conferences or after re-resection with TURb were compiled. The validation was performed by junior doctors with experience in urological research under the supervision of an experienced urologist (A.S.), who was frequently consulted.

As previously mentioned, pre-, peri-, and post-operative complications have been registered in a separate form in the SNRUBC since 2011 [4]. In the present study, the form was referred to as detailed cystectomy data. Previously, the clinical tumour classification was only recorded in the initial registration form in the SNRUBC. The unvalidated cT-category was defined as the highest cT-category recorded in the SNRUBC (pre-cystectomy), regardless of being noted in the initial registration form or in the detailed cystectomy data.

By reviewing each patient’s medical journal, relevant medical data were compiled in a database, such as age, sex, Charlson-Age Comorbidity Index (CACI), pathological classification of primary tumour (pT-category), and information regarding neoadjuvant chemotherapy or induction chemotherapy. The registered data in the SNRUBC (unvalidated data) were not blinded for the reviewer, but no analysis was performed before the validation process was completed.

Pathological reports regarding the primary TURB specimens and the cystectomy specimens were used for the pathological tumour classification (pT). No central pathological review was performed. Concomitant carcinoma in situ was noted for most non-muscle invasive tumours, while no reliable documentation in muscle-invasive tumours was made. Variants of histology (micropapillary, plasmacytoid, sarcomatoid, small cell, or neuroendocrine) were not reliably documented in the pathology reports.

TIBD was confirmed by reviewing the cystoscopy report together with computed tomography examinations pre-cystectomy. Only UBC within a bladder diverticulum confirmed by visual findings on cystoscopy together with radiologically confirmed diverticula on computed tomography was included. Based on the cystoscopy report, all suspected or visually confirmed cases of TIBD were compiled in a separate list. Furthermore, TAH was assessed by reviewing computed tomography urography (CTU) statements pre-cystectomy. These statements were generally confirmed by two separate radiologists, and all patients with radiologically suspected dilation of the renal pelvis or confirmed hydronephrosis were compiled in a separate list. All suspected cases of TAH and TIBD had CTU re-reviewed by one experienced radiologist (E.E.). The re-review was not performed blinded with regard to the initial radiological statement. Only clinically significant hydronephrosis caused by a tumour located near the urethral orifice was classified as TAH. Patients with cT1 in BD, confirmed by this CTU re-review, were registered and compared as cT2 in the patient-per-patient registration, but described as a special group, T2/T3, as the determination of the exact cT-category might be difficult. Similarly, patients with cT1 and cT2 with radiologically and re-evaluated confirmed hydronephrosis were registered as cT3. Patients with both TIBD and TAH were registered as cT3. TaG2 tumours with concomitant Tis were registered as cTis.

The primary outcome was the discrepancy between the unvalidated cT-category registered in the SNRUBC and the validated cT-category in our study. A comparison between the validated cT-categories and pT-categories in patients that did not receive chemotherapy was also made as a subgroup analysis.

### Statistical Methods

Descriptive statistics concerning patient characteristics and clinical tumour stages were studied in the groups with and without discrepancy between cT registered in the SNRUBC and cT determined after validation. Interval level data are presented with the mean (SD) or median (IQR) for each allocated group using two independent sample *t*-tests or Mann–Whitney U-tests. Categorical variables are described with frequency tables and groups compared using the Chi-squared test (Chi2-test). Unadjusted and adjusted logistic regression were used to evaluate variable association with discrepancy between registered cT (unvalidated) and the re-registered cT (validated). The results are reported as the odds ratios (ORs) with the corresponding 95% confidence intervals (CIs). A significance level of 0.05 was used. Statistical analyses were performed using IBM SPSS Statistics, Version 29 (IBM Corporation, Armonk, NY, USA).

## 3. Results

In total, 295 patients with UBC of urothelial origin underwent RC (Figure 1), of whom 233 had muscle-invasive disease (cT2-cT4b) and 121 patients were radiologically re-reviewed. Three patients had concomitant cancer in situ in the urethra; however, none of these tumours were regarded as cT4a. There was a significant discrepancy between the cT-category registered in the SNRUBC (unvalidated) and the cT-category in the re-registration (validated). The discrepancy between validated and unvalidated data was found in 87 of the patients (29.5%) (Table 1). In 10 cases, a higher cT-category was found in the SNRUBC compared to the re-registration, and in 77 cases, a lower cT-category was found in the SNRUBC. In total, 146 patients received chemotherapy; 136 of these received neoadjuvant chemotherapy (NAC), and 10 received induction chemotherapy (IC). There were no statistical differences between the cT discrepancy groups regarding age, sex, CACI, and NAC. In total, 70 patients (23.7%) had confirmed TAH, and 48 of them (55.2%) showed discrepancy regarding the cT-category. Furthermore, 23 patients (7.8%) had confirmed TIBD, and of these, 12 (52.2%) showed discrepancy regarding the cT-category. An overlap of TAH and TIBD was found in six cases (2.0%), and three (50%) of them were upstaged.

There was a large difference in the distribution between the cT-categories registered in the SNRUBC and the validated cT-categories, especially regarding MIBC (Table 2). In a logistic regression model (adjusted for the other possible predictor variables), TIBD and TAH were two significant factors impacting the discrepancy between cT-categories (Table 3). The analyses regarding TIBD and TAH showed an odds ratio (OR) of 7.71, a 95% confidence interval (CI) of 2.53–23.6, *p* < 0.001 and an OR of 17.7, a 95% CI of 8.53–36.8, *p* < 0.001, respectively. In the same analysis, the occurrence of detailed data regarding cystectomy, as well as overlapping TIBD and TAH were not associated with discrepancy between cT-categories.

In 32.9% of the patients that did not receive chemotherapy, cT was equal to pT (post-cystectomy), and in 27.5%, pT was upstaged compared to cT, while in 39.6%, pT was downstaged compared to cT (Table 4). Most of all discrepancies, 53 cases (61%), were found during the years 2009–2012, when the proportion of patients with detailed data regarding cystectomy was low (32%) (Figure 2). During the later years (2013–2020), this proportion substantially increased. When considering the full study period, 75% of all patients registered in the SNRUBC had detailed data regarding cystectomy (Table 1).

## 4. Discussion

In the present study, the registered cT-categories underestimated the true T categories partially due to a lack of consideration of the radiological examination. The validation of the cTs registered in the SNRUBC demonstrated an incorrect registration in 87 cases (29.5%), of which 77 resulted in upstaging of the tumour. In 60 (70%) out of these 87 patients, the misclassification was due to a lack of consideration of radiological examination, as tumours were reclassified after a thorough radiological review of all patients with suspected TAH or TIBD. In some cases, cTa was registered in the SNRUBC when the highest cT-category should be cTis, explaining the decrease of cTa tumours (unvalidated) and the increase in cTis after validation.

Using data in the SNRUBC, a suspected misclassification was previously presumed by Russell et al., who studied survival regarding NAC for MIBC. This study reported a high incidence of cT2 tumours in the SNRUBC, cT2 tumours comprising 86% of the total registered MIBCs during the years 2008–2014 [20]. In the present study considering only MIBC, the proportion of T2 tumours changed from 77% registered in the SNRUBC to 60% after validation. These results are in line with those of Russel et al. [20] and emphasise the need for an optimisation of the TNM classification in all bladder cancer registers.

In this study cohort, the presence of tumour-associated hydronephrosis was common (23.7%). This often resulted in upstaging of cT1 and cT2 to cT3, which is in accordance with clinical urological practice. The presence of TAH, therefore, was a significant predictor regarding discrepancy in the adjusted logistic regression analysis, OR of 17.7, *p* < 0.001. It could partially explain the decrease in the percentage of cT1 from 16.9% (unvalidated) to 10.8% (validated) and the increase of cT3 from 11.2% to 26.8%, respectively. A meta-analysis with 10 461 patients showed a significant correlation between hydronephrosis and advanced disease and a poorer survival rate [21]. Bartsch et al. found a similar association between hydronephrosis and advanced tumours. Additionally, they found hydronephrosis to be an individual prognostic factor for survival [8]. Thus, TAH is an essential factor that should not be omitted in the SNRUBC registration.

UBC in BD is a common cause of pathologic upstaging, as observed by Voskuilen et al. [14], who found upstaging of the primary TIBD to that of at least pT2 in 55% of the patients with cTa, cTis, or cT1. Hu et al. found that 42% of the patients with UBC in BD had pT2 [22]. The results of those studies indicated inaccuracy in the clinical staging of patients with TIBD. In the present study, 7.8% of the patients had a TIBD, leading to discrepancies for cT staging. This is a relative high incidence compared to other publications [12]. The validation often resulted in upstaging of cT1 in BD to cT2 (and considered as T2/T3), contributing to discrepancy between the SNRUBC and the validated data. Adjusted logistic regression analysis found TIBD to be a significant predictor for incorrect registration, OR of 7.71, *p* < 0.001. Idrees et al. stated that there is no pT2 in BD; instead, pT1 should be considered as pT3 [23]. However, even if this study had followed this recommendation, the discrepancy in the subgroup cT1 would remain substantially similar. Here, we classified TIBD as cT2 when comparing discrepancy, but noted such tumours as T2/T3, highlighting the difficulty of this tumour classification. Overlapping TIBD and TAH must also be taken into consideration. The interaction between TAH and TIBD had a low OR, 0.03, *p* = 0.002, meaning that overlapping TAH and TIBD is easier to classify than the combined effect of TAH and TIBD suggests.

In the original radiological statement, CTU was most often assessed by two independent radiologists to evaluate the urinary tract, including TAH and TIBD, as part of clinical practice. In the present study, an additional radiologist reviewed the images confirming TAH and/or TIBD. This strengthened the reliability of our proposed theory regarding factors contributing to incorrect registration. Further, in nearly 30% of the patients that did not receive chemotherapy, pT was upstaged compared to cT. This further supports the observation that TAH and TIBD often might be upstaged and carry a bad prognosis.

Timely diagnosis with correct cT-classification and NAC is important for the clinical outcome after RC [24]. Positive surgical margins at the time of RC are generally associated with poor outcome with high recurrence and mortality rates [25]. Optimised cT-classification can, therefore, theoretically minimise risk for positive surgical margins at the time of RC. However, in this study, positive surgical margins after RC were not thoroughly investigated, as a central pathological review was not performed. Further treatment and survival analyses are needed to address this research question.

In 2011, detailed data regarding cystectomy were added to the SNRUBC, and the proportions of incorrect registration substantially decreased thereafter. This is supported by the OR of less than 1 in the logistic regression analysis, OR of 0.16 (95% CI, 0.08–0.32), *p* < 0.001. Not having an updated cT-category in the SNRUBC prior to cystectomy was most likely a large contributing factor to discrepancy, especially regarding NMIBC. This updated form regarding data prior to cystectomy was probably the most-important factor for the increase of more-accurate registration during the years 2013–2020 (Figure 2). Still, the registration in the SNRUBC is seldomly carried out by urologists. This could also be a contributing factor for the differences between the registered cT in the SNRUBC and the cT validated in this study, indicating a need for more involvement by urologists in Sweden in the registration process.

The present study was not without limitations. Firstly, the relatively small study cohort of 295 patients was a main limitation. There was a risk of excessive emphasis on TAH and TIBD as factors contributing to discrepancy with the relatively small cohort. Secondly, the study was conducted retrospectively, and there was a risk of bias when the data were selected and analysed with this approach. The validation was performed by junior doctors and not by urologists, but under the direct supervision of an experienced urologist. During the 11-year period (2009–2020), many changes have been made regarding the routines for the registration of the data in the SNRUBC. This must be considered when examining incorrect registration. During the last few years, better manuals for registration have been implemented. Multi-disciplinary conferences regarding treatment have been more common during the years studied, which was not considered when discussing discrepancy. Similarly, the number of patients re-resected with TURb was not investigated. Concomitant CIS at TURb constitutes a risk for worse clinical outcomes [26]. In this study, the presence of concomitant CIS was not thoroughly investigated, as this variable is also not registered in the SNRUBC. The present study was limited to urothelial carcinoma and did not include all histological subtypes of UBC. The presence of variant histologies of urothelial carcinoma and the impact on possible discrepancies were not investigated. A total of 18 patients (Figure 1) with other cancer types or other histological subtypes than urothelial carcinoma was excluded. The impact of these factors with regard to discrepancy between the SNRUBC and the cT-category determined after validation should be investigated in the future. More centres need to be included to fully investigate the extent of discrepancies in the SNRUBC, especially considering the possibility of varying accuracy in clinical and pathological classification regarding variant histologies [18]. BP is also an important part of the diagnostic procedure in UBC [27,28]. According to the literature, not all patients undergo BP during TURb [29,30]. For that reason, the present study excluded BP as a studied variable, although a palpable immobile mass on BP has been reported to indicate the cT4-category [29]. The exclusion of BP might have affected the outcomes in this study, to a limited extent. This study validated the cT-categories, but not the cNM-categories at all. It is of great clinical importance that the entire registered data be validated in the future, and ideally, such validation should include radiological examination. However, the result of this study can be used as a basis for a larger validation study.

## 5. Conclusions

The result of this study demonstrated that the registered data regarding the cT-category in the SNRUBC are not completely reliable, especially during the years 2009–2012. The treatment of UBC is performed based on the cTNM-category; it is, therefore, crucial to classify the tumour correctly for accurate register quality. Considering radiological findings such as tumour-associated hydronephrosis and tumour in bladder diverticulum is an essential part to improve the classification. We propose the implementation of a clearer manual regarding the inclusion of TAH and TIBD when registering cT-categories in the SNRUBC. A larger, multicentre study is needed to fully investigate the extent of misclassification in the SNRUBC.

## Figures and Tables

**Figure 1 jpm-13-01163-f001:**
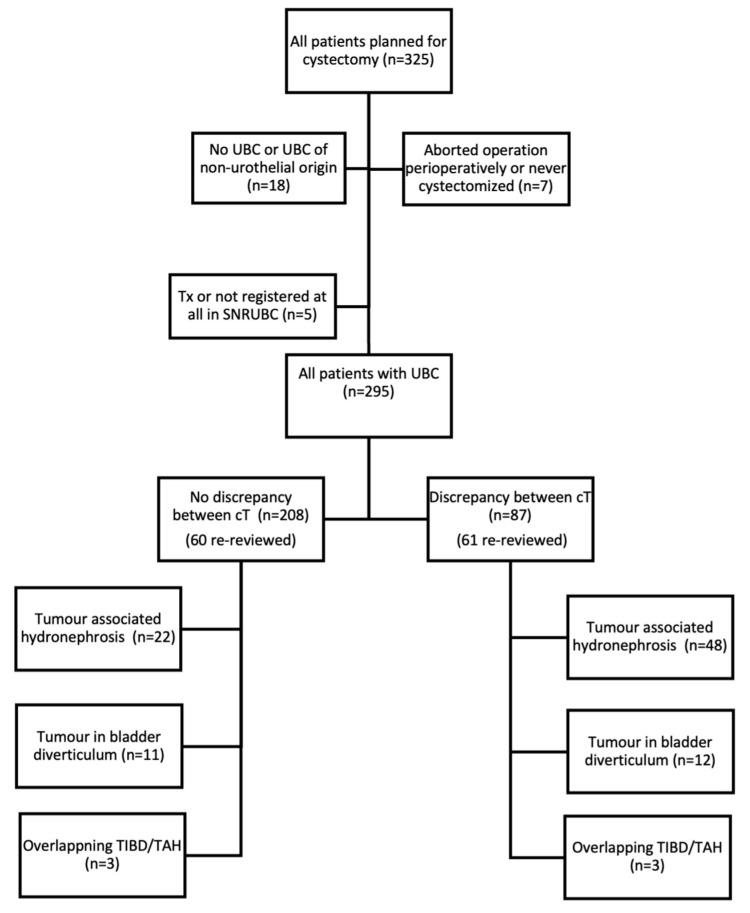
Flow chart of inclusion and results of all evaluated patients. UBC: urinary bladder cancer. SNRUBC: Swedish National Register of Urinary Bladder Cancer. Tx: there is insufficient information for stage classification. cT: clinical classification of the primary tumour. TIBD: tumour in bladder diverticulum. TAH: tumour-associated hydronephrosis.

**Figure 2 jpm-13-01163-f002:**
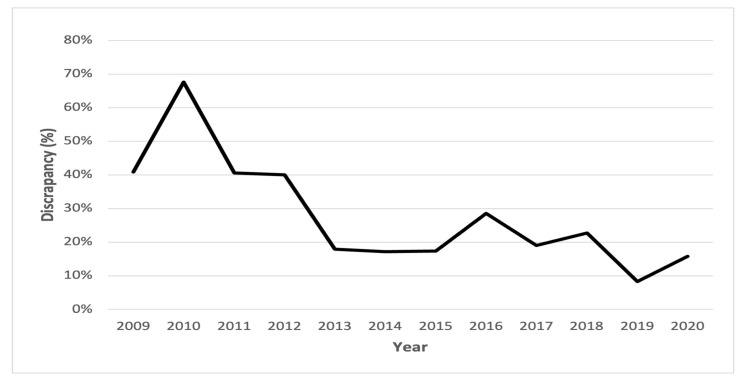
Percent discrepancy in cT-categories between data in the SNRUBC and the re-registration per year. cT: clinical classification of the primary tumour.

**Table 1 jpm-13-01163-t001:** Patient characteristics and comparisons between the group with discrepancy regarding cT-category and the group without discrepancy.

	Discrepancy between cTs		Total	*p*-Value
	No, n = 208	Yes, n = 87	n = 295	
Age, mean (SD)	69.4 (7.70)	69.6 (8.40)	69.4 (7.90)	0.416 ^†^
Sex, n (%)				0.73 *
Male	164 (78.8)	67 (77.0)	231 (78.3)	
Female	44 (21.2)	20 (23.0)	64 (21.7)	
Detailed cystectomy data, n (%)				<0.001 *
Yes	175 (84.1)	47 (54.0)	222 (75.3)	
No	33 (15.9)	40 (46.0)	73 (24.7)	
Re-evaluated radiologically, n (%)				<0.001 *
No	148 (71.2)	26 (29.9)	174 (59.0)	
Yes	60 (28.8)	61 (70.1)	121 (41.0)	
cT-category, n (%)				<0.001 *
Ta	12 (5.8)	0 (0.0)	12 (4.1)	
Tis	12 (5.8)	6 (6.9)	18 (6.1)	
T1	29 (13.9)	3 (3.4)	32 (10.8)	
T2	119 (57.2)	20 (23.0)	139 (47.1)	
T3	26 (12.5)	53 (60.9)	79 (26.8)	
T4a	10 (4.8)	2 (2.3)	12 (4.1)	
T4b	0 (0.0)	3 (3.4)	3 (1.0)	
CACI, mean (SD)	5.10 (1.20)	5.10 (1.20)	5.10 (1.2)	0.47 ^†^
NAC				0.43 *
No	109 (52.4)	50 (57.5)	159 (53.9)	
Yes	99 (47.6)	37 (42.5)	136 (46.1)	
IC				0.15 *
No	203 (97.6)	82 (94.3)	285 (96.6)	
Yes	5 (2.4)	5 (5.7)	10 (3.4)	
Chemotherapy type, n (%)				0.53 *
CarboGem	2 (1.0)	1 (1.1)	3 (1.0)	
HD-MVAC	91 (43.8)	32 (36.8)	123 (41.7)	
HD-MVEC	8 (3.8)	7 (8.0)	15 (5.1)	
Other	3 (1.4)	2 (2.3)	5 (1.7)	
None	104 (50.0)	45 (51.7)	149 (50.5)	
Tumour in bladder diverticulum, n (%)				0.013 *
No	197 (94.7)	75 (86.2)	272 (92.2)	
Yes	11 (5.3)	12 (13.8)	23 (7.8)	
Tumour-associated hydronephrosis, n (%)				<0.001 *
No	186 (89.4)	39 (44.8)	225 (76.3)	
Yes	22 (10.6)	48 (55.2)	70 (23.7)	
Overlapping TIBD/TAH, n (%)				0.27 *
No	205 (98.6)	84 (96.6)	289 (98.0)	
Yes	3 (1.4)	3 (3.4)	6 (2.0)	
S-creatinine pre-cystectomy, median (IQR)	80.0 (71–98.5)	94.0 (76–109)	84.0 (72–102)	0.011 º
Nephro-pyelostomy/stent pre-cystectomy, n (%)				<0.001 *
No	182 (87.5)	61 (70.1)	243 (82.4)	
Yes	26 (12.5)	26 (29.9)	52 (17.6)	

cT-category: clinical staging of the primary tumour. Detailed cystectomy data: data in the SNRUBC regarding cT-category in conjunction with cystectomy. CACI: Aged-adjusted Charlson Comorbidity Index. NAC: neoadjuvant chemotherapy. IC: induction chemotherapy. TIBD: tumour in bladder diverticulum. TAH: tumour-associated hydronephrosis. * = tested with Chi2test. ^†^ = tested with *t*-test. º = tested with Mann–Whitney U-test.

**Table 2 jpm-13-01163-t002:** cT-categories, unvalidated and validated, n (%).

	cT Validated	Ta	Tis	T1	T2	T3	T4a	T4b	Total (%)
cT Unvalidated									
Ta		12	5	0	1 [1]	6	0	0	24 (8.1)
Tis		0	12	3	1 [1]	0	0	0	16 (5.4)
T1		0	1	29	12 [4]	8	0	0	50 (16.9)
T2		0	0	0	119 [11]	36 [4]	1	2	158 (53.6)
T3		0	0	0	5 [2]	26	1	1	33 (11.2)
T4a		0	0	0	1	1	10	0	12 (4.1)
T4b		0	0	0	0	2	0	0	2 (0.7)
Total (%)		12 (4.1)	18 (6.1)	32 (10.8)	139 (47.1)	79 (26.8)	12 (4.1)	3 (1.0)	295 (100)

cT: clinical classification of the primary tumour. The distribution of all TIBCs is noted with square brackets []. Tumour in bladder diverticulum (n), denoted as T2, but considered to be T2/T3.

**Table 3 jpm-13-01163-t003:** Factors impacting cT stage discrepancy analysed with logistic regression, unadjusted (tested individually) and adjusted (tested adjusted for all included predictor variables).

Predictors				
	Unadjusted		Adjusted	
	OR (95% CI)	*p*-Value	OR (95% CI)	*p*-Value
Tumour-associated hydronephrosis	10.4 (5.65–19.2)	<0.001	17.7 (8.53–36.8)	<0.001
Tumour in bladder diverticulum	2.87 (1.21–6.77)	0.016	7.71 (2.53–23.6)	<0.001
Overlapping TIBD/TAH	2.44 (0.48–12.3)	0.28	0.03 (0.00–0.29)	0.002
Detailed cystectomy data	0.22 (0.13–0.39)	<0.001	0.16 (0.08–0.32)	<0.001

TAH: tumour-associated hydronephrosis. TIBD: tumour in bladder diverticulum. Cystectomy data: data in the SNRUBC regarding cT-category in conjunction with cystectomy.

**Table 4 jpm-13-01163-t004:** cT-category compared to pT-category (post-cystectomy) for patients that did not receive chemotherapy (NAC or IC).

cT-Category	cT > pT n (%)	cT < pT n (%)	cT = pT n (%)	Total n
Ta/Tis	13 (44.8)	2 (6.9)	14 (48.3)	29
T1	16 (53.3)	7 (23.3)	7 (23.3)	30
T2	17 (28.8)	26 (44.1)	16 (27.1)	59
T3	10 (37.0)	6 (22.2)	11 (40.7)	27
T4a/T4b	3 (75.0)	0 (0.0)	1 (25.0)	4
Total	59 (39.6)	41 (27.5)	49 (32.9)	149

cT: clinical classification of the primary tumour. pT: pathological classification of the primary tumour. NAC: neoadjuvant chemotherapy. IC: induction chemotherapy.

## Data Availability

Upon reasonable request, the corresponding author can make available all codified data from the clinical database used for this study.

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
