# Peer review of "A Validation Study of cT-Categories in the Swedish National Urinary Bladder Cancer Register—Norrland University Hospital"

_jpm, 2023, doi:10.3390/jpm13071163_

Round 1

Reviewer 1 Report

The manuscript by Wiberg E. et al entitled "a validation study of cT-categories in the Swedish national urinary bladder cancer register- Norrland University Hospital" is nicely-written and sheds a light on a very important issue that tackles discrepancy in data gathering from different modalities and the subsequent interpretation. The paper is sound and my only comment is to fix the English language as there are some editings that need to be done.

Moderate editing of English language required

Author Response

We thank the Reviewer for comments below and present our responses accordingly:

Comments and Suggestions for Authors

The manuscript by Wiberg E. et al entitled "a validation study of cT-categories in the Swedish national urinary bladder cancer register- Norrland University Hospital" is nicely-written and sheds a light on a very important issue that tackles discrepancy in data gathering from different modalities and the subsequent interpretation. The paper is sound and my only comment is to fix the English language as there are some editings that need to be done.

Comments on the Quality of English Language

Request: Moderate editing of English language required

Reply: We have performed an editing of the English language.

Reviewer 2 Report

In this study the authors investigated bladder cancer cT stages among pts treated with radical cystectomy within a registry-based setting aiming to determine the incidence of misclassification of the cT-staging. A sensitivity analysis on tumors associated with hydronephrosis and/or with diverticulum involvement was performed. Overall interesting results.

Please pay attention to the following comments for major revision.

As correctly mentioned in the Introduction, urothelial carcinoma (pure) is the most common histological subtypes in this specific spectrum. As the main focus of the paper is related to misclassification considering clinical categories it must be pointed out the relevance of the variant histologies (subtypes histology as per novel WHO 2022) as a recognized drivers of biological aggressiveness and prognosis. Moreover, variant histologies have been historically correlated with aggressive disease in term of both clinical and pathological stage. The importance of these entities is worth mentioning. Recently, two major studies have been conducted in this scenario: NMIBC - doi: 10.1007/s00428-021-03264-6, MIBC - doi: 10.1111/bju.15984. Furthermore, these findings are even more worth mentioning considering the variability between centers (doi: 10.1016/j.urolonc.2022.01.008). No information about variant - subtypes - histology have been reported in this paper. The lack of these data is one of the main limitations of the current report. Could the authors provide these information? Otherwise they need to highlight these lack in the appropriate section.

Another point of discussion could be related to the surgical margins after RC. As the cT-category could prompt the treatment towards a more intense regimen of preoperative cares, consolidative surgery have the goal to reach oncological radically. Could a more refined stratification - based on your findings - be part of a standardized approach to minimize positive surgical margins at time of RC? No information about the margin status were reported. Could the authors show these data? The impact of surgical margins status and location may impact cancer-specific survival after RC. Two major reports recently highlight such aspects (doi: 10.3390/cancers14235740, doi: 10.1007/s00345-021-03776-5). The authors should further discuss these findings.

Any information about the presence of concomitant CIS?

The sensitivity analysis considering tumor associated hydronephrosis and/or diverticulum involvement included almost 10% of the final studied cohort. Check concordance between Tables and Main Text. Thus related findings, have to be softened according to the small number of events. Please highlight this in the limitation paragraph.

295 pts initially included: please stress the small sample size as one of the main limitation.

Author Response

We thank the Reviewer for comments below and present our responses accordingly:

Comments and Suggestions for Authors

In this study the authors investigated bladder cancer cT stages among pts treated with radical cystectomy within a registry-based setting aiming to determine the incidence of misclassification of the cT-staging. A sensitivity analysis on tumors associated with hydronephrosis and/or with diverticulum involvement was performed. Overall interesting results.

Please pay attention to the following comments for major revision.

As correctly mentioned in the Introduction, urothelial carcinoma (pure) is the most common histological subtypes in this specific spectrum. As the main focus of the paper is related to misclassification considering clinical categories it must be pointed out the relevance of the variant histologies (subtypes histology as per novel WHO 2022) as a recognized drivers of biological aggressiveness and prognosis. Moreover, variant histologies have been historically correlated with aggressive disease in term of both clinical and pathological stage. 

Request: The importance of these entities is worth mentioning. Recently, two major studies have been conducted in this scenario: NMIBC - doi: 10.1007/s00428-021-03264-6, MIBC - doi: 10.1111/bju.15984. Furthermore, these findings are even more worth mentioning considering the variability between centers (doi: 10.1016/j.urolonc.2022.01.008). 

Reply: We have now added these references, please see ref [15] and [16]

Request: No information about variant - subtypes - histology have been reported in this paper. The lack of these data is one of the main limitations of the current report. Could the authors provide these information? Otherwise they need to highlight these lack in the appropriate section.

Reply: Unfortunately we cannot provide this information, but we acknowledge the noted importance of histological variant subtypes. In order to highligt this matter we have now mentioned this in the INTRODUCTION on lines 63-69 and also shortly in the DISCUSSION on lines 296-297.

Request: Another point of discussion could be related to the surgical margins after RC. As the cT-category could prompt the treatment towards a more intense regimen of preoperative cares, consolidative surgery have the goal to reach oncological radically. Could a more refined stratification - based on your findings - be part of a standardized approach to minimize positive surgical margins at time of RC? No information about the margin status were reported. Could the authors show these data? The impact of surgical margins status and location may impact cancer-specific survival after RC. Two major reports recently highlight such aspects (doi: 10.3390/cancers14235740, doi: 10.1007/s00345-021-03776-5). The authors should further discuss these findings.

Reply: We certainly acknowledge the importance of radical margins and the request from the Reviewer is clearly motivated. Yet, we do not have the data requested. To emhasize this matter we have added text on lines 262-269 in the DISCUSSION and also added the suggested references [24] and [25].

Request: Any information about the presence of concomitant CIS?

Reply: As the Reviewer points out, concomitant CIS in MIBC-patients is a valuable factor. In lines 292-295 in the DISCUSSION we have added following text plus reference [26]: Concomitant CIS at TURb constitutes a risk for worse clinical outcomes [26]. In this study, presence of concomitant CIS has not been thoroughly investigated, as this variable is also not registered in the SNRUBC.

Request: The sensitivity analysis considering tumor associated hydronephrosis and/or diverticulum involvement included almost 10% of the final studied cohort. Check concordance between Tables and Main Text. Thus related findings, have to be softened according to the small number of events. Please highlight this in the limitation paragraph.295 pts initially included: please stress the small sample size as one of the main limitation.

Reply: We do agree with the Reviewer, the concordance is checked and we have added following text in the DISCUSSION (lines 281-283):

The present study is not without limitations. Firstly, the relatively small study cohort of 295 patients is a main limitation. There is a risk of excessive emphasis on TAH and TIBD as factors contributing to discrepancy with the relatively small cohort.

Round 2

Reviewer 2 Report

The authors revised the manuscript fully and properly.